# Ectopic Overexpression of Pineapple Transcription Factor AcWRKY31 Reduces Drought and Salt Tolerance in Rice and *Arabidopsis*

**DOI:** 10.3390/ijms23116269

**Published:** 2022-06-03

**Authors:** Youmei Huang, Fangqian Chen, Mengnan Chai, Xinpeng Xi, Wenhui Zhu, Jingang Qi, Kaichuang Liu, Suzhuo Ma, Han Su, Yaru Tian, Huiying Zhang, Yuan Qin, Hanyang Cai

**Affiliations:** 1College of Life Sciences, Fujian Provincial Key Laboratory of Haixia Applied Plant Systems Biology, State Key Laboratory of Ecological Pest Control for Fujian and Taiwan Crops, Fujian Agriculture and Forestry University, Fuzhou 350002, China; hym9995@163.com (Y.H.); chenfangqian96@163.com (F.C.); chaimengnan1@163.com (M.C.); 18235795238@163.com (X.X.); 3200130066@fafu.edu.cn (W.Z.); 1200102024@fafu.edu.cn (J.Q.); vegetarianliu@163.com (K.L.); 13703419836@163.com (S.M.); su1135532091@163.com (H.S.); eryntian9911@126.com (Y.T.); zhanghuiying_1988@163.com (H.Z.); yuanqin@fafu.edu.cn (Y.Q.); 2State Key Laboratory for Conservation and Utilization of Subtropical Agro-Bioresources, Guangxi Key Lab of Sugarcane Biology, College of Agriculture, Guangxi University, Nanning 530004, China; 3Pingtan Science and Technology Research Institute, Fujian Agriculture and Forestry University, Pingtan 350400, China

**Keywords:** pineapple, wrky transcription factors, *AcWRKY31*, RNA-seq, drought, salt

## Abstract

Pineapple (*Ananas comosus* (L.) Merr.) is an important tropical fruit with high economic value, and its growth and development are affected by the external environment. Drought and salt stresses are common adverse conditions that can affect crop quality and yield. WRKY transcription factors (TFs) have been demonstrated to play critical roles in plant stress response, but the function of pineapple WRKY TFs in drought and salt stress tolerance is largely unknown. In this study, a pineapple *AcWRKY**31* gene was cloned and characterized. *AcWRKY31* is a nucleus-localized protein that has transcriptional activation activity. We observed that the panicle length and seed number of *AcWRKY31* overexpression transgenic rice plants were significantly reduced compared with that in wild-type plant ZH11. RNA-seq technology was used to identify the differentially expressed genes (DEGs) between wild-type ZH11 and *AcWRKY31* overexpression transgenic rice plants. In addition, ectopic overexpression of *AcWRKY31* in rice and *Arabidopsis* resulted in plant oversensitivity to drought and salt stress. qRT-PCR analysis showed that the expression levels of abiotic stress-responsive genes were significantly decreased in the transgenic plants compared with those in the wild-type plants under drought and salt stress conditions. In summary, these results showed that ectopic overexpression of *AcWRKY31* reduced drought and salt tolerance in rice and *Arabidopsis* and provided a candidate gene for crop variety improvement.

## 1. Introduction

Plants are often exposed to various environmental stresses during all stages of plant growth and development. Abiotic stresses such as drought, salinity, high temperature and cold can affect plant growth and development, yield, and quality [1]. In order to survive these adverse conditions, plants have developed complex defense mechanisms to face external environmental changes. Transcription factors (TFs) play crucial roles in plant defense systems by regulating genes expression. The WRKY transcription factor family is one of the largest transcription factor families in higher plants, and various family members play important roles in many aspects of physiological processes and resist the adverse environment [2,3,4,5].

A typical WRKY transcription factor contains the 60-amino-acids-long WRKY domain, which has the highly conserved signature WRKYGQK in the *n*-terminal, followed by a C2H2- or C2HC-type of the zinc-finger motif [6]. In some WRKY TFs, the WRKY residues in the WRKY domain are replaced by WRRY, WSKY, WKRY, WVKY and WKKY residues [7,8]. In rice, the WRKY family has multiple WRKY domain variants, of which WRKYGEK and WRKYGKK are the most common [7]. Both the heptapeptide sequence and the zinc-finger motif are necessary for the binding of WRKY TFs to the consensus W-box cis-elements. Based on the number of WRKY domains and the structure of the zinc-finger motif, the WRKY TFs can be classified into three main groups (I-III). Group I WRKY TFs contain two WRKY domains and a C2H2-type zinc-finger motif, and the DNA binding function of this kind of WRKY transcription factor is mainly mediated by the C-terminal WRKY domain, while the WRKY TFs in Group II contain only one WRKY domain, and its zinc finger structure is also C2H2 type. In addition, the sequence of the WRKY domain in Group II is more similar to that of the C-terminal WRKY domain of Group I WRKY transcription factors. Group III members have one WRKY domain with a C2HC-type zinc-finger motif [6,7,9]. In addition, in some higher plants, the members of Group II can be divided into IIa, IIb, IIc, IId and IIe subgroups according to the main amino acid sequence composition [7].

Since the first WRKY gene *SPF1* was identified from sweet potato [10], the functions of WRKY proteins have been well studied from various plant species. Substantial evidence has shown that WRKY TFs are involved in many different biological processes. For example, in the process of seed germination, *AtWRKY27* is a negative regulator in the GA signal pathway [11]. *AtWRKY2* mediates seed germination and post germination arrest of growth and development through the ABA pathway [12]. Loss of *AtWRKY53* can delay the senescence of plants, while overexpression of *AtWRKY53* will promote the senescence of plants [13]. Furthermore, a large number of *WRKY* genes take part in the regulation of plant tolerance to abiotic stress. In *Arabidopsis*, *WRKY39* genetically interacts with SA- and JA-signaling pathways to positively regulate response to heat stress, and overexpression of *WRKY39* can improve plant tolerance to heat stress [14]. *VvWRKY11* from grapevine is involved in the response to dehydration stress [15]. Overexpression of *OsWRKY45* and *OsWRKY72* in rice can significantly improve plant tolerance to drought and salt stress [16,17]. Under the control of the rice HSP101 promoter, overexpression of *OsWRKY11* will enhance the tolerance of heat and drought stresses of transgenic rice seedlings [18].

Pineapple (*Ananas comosus* (L.) Merr.) is a perennial monocotyledonous plant, which is a famous tropical fruit. Pineapple grows in South America and now in almost all tropical and subtropical regions in the world. It has been widely planted in 85 countries [19]. However, environmental stress seriously affects the growth and development of pineapple plants. Pineapple fruits will be sunburned under high temperature, and low temperature will slow down the growth of the plants, thus affecting the quality and yield of pineapple [20]. Because pineapple has high nutritional and economic value, the identification of its important functional genes has also aroused great interest. With the completion of pineapple genome sequencing, the analysis of multiple pineapple gene families, including WRKY transcription factors, has been reported [20,21,22,23,24]. At present, 54 members of the pineapple WRKY transcription factor family have been identified [20], but there are few reports on the function of pineapple WRKY transcription factors.

WRKY transcription factor is one of the most important transcription factors in plants [2]. However, there are few reports on the function of the *WRKY* gene in pineapple. It has been reported that the expression of *AcWRKY31* (*Aco000358.1*) was induced under cold and drought stress conditions [20], but the function of *AcWRKY31* in still limited. In order to investigate the role of *AcWRKY31* in plant growth and development, *AcWRKY31* was identified and cloned from the pineapple genome. A comprehensive analysis, including multiple sequence alignment, subcellular localization and transcriptional activation activity analysis, was performed. Due to the immature genetic transformation system of pineapple, it is still difficult to study the function of *AcWRKY31* in pineapple. Since both rice and *Arabidopsis* are important model plants, we constructed *AcWRKY31* overexpression transgenic plants to further investigate the function of *AcWRKY31*. In addition, we provide evidence that overexpression of *AcWRKY31* in rice and *Arabidopsis* will affect plant growth and development and change the expression levels of multiple stress-responsive genes under drought and salt stresses. This is the first work showing the role of pineapple AcWRKY31 transcription factor in plant growth and development and stress response. This study laid a theoretical foundation for the functional research of *AcWRKY31* and provided new genes and new ideas for crop breeding and improvement in both dicots and monocots.

## 2. Results

### 2.1. Sequence Alignment of AcWRKY31 

AcWRKY31 (Aco000358.1) and its homologous protein sequences were downloaded from Phytozome13. Multiple alignments showed that AcWRKY31 showed high sequence similarity with AcWRKY25 (Aco005520.1), OsWRKY113 (LOC_Os06g06360.1) and AtWRKY53 (AT4G23810.1). AcWRKY31 and its homologous proteins contained a highly conserved WRKY domain and C2HC-type of the zinc-finger motif (Figure 1). 

### 2.2. Subcellular Localization and Transcriptional Activity of AcWRKY31

In order to investigate the subcellular location of AcWRKY31, the coding sequence of *AcWRKY31* was fused to the N-terminal of the green fluorescent protein (GFP) under the control of the CaMV 35S promoter, generating a fusion protein vector *35S::AcWRKY31::GFP*. Then, the recombinant vector was transformed into *Nicotiana benthamiana* leaves, and the empty vector *35S::GFP* was used as control. Microscopic visualization showed that AcWRKY31::GFP was exclusively localized in the nucleus, whereas the control GFP signal was observed in the whole cell including the cell membrane and nucleus (Figure 2A). These results suggested that AcWRKY31 is a nucleus-localized protein.

To investigate the transcriptional activation activity of AcWRKY31, a yeast assay system was used. The full-length coding DNA sequence (CDS) of *AcWRKY31* was cloned into the pGBKT7 vector. Then, we transferred pGBKT7-AcWRKY31 + pGADT7-T into the AH109 yeast strains, and the yeast strains containing pGBKT7 or pGBKT7-AcWRKY31 were used as control. As shown in Figure 2B, the yeast strains containing pGBKT7-AcWRKY31 + pGADT7-T grew normally on the SD/-Trp/-Leu medium and turned blue on the SD/-Trp/-Leu/-His/X-α-gal medium. The growth of the yeast cells harboring pGBKT7-AcWRKY31 were inhibited on the SD/-Trp/-Leu medium and SD/-Trp/-Leu/-His/X-α-gal medium. However, the yeast cells transformed with pGBKT7 could not grow on the SD/-Trp/-Leu medium and SD/-Trp/-Leu/-His/X-α-gal medium (Figure 2B). These results indicate that AcWRKY31 has transcriptional activation activity.

### 2.3. Expression Profiles of Pineapple AcWRKY31 Response to Various Abiotic Stresses

Several studies have reported that some WRKY TFs were involved in abiotic stress response [3,4]. To investigate the role of AcWRKY31 in abiotic stress response, the one-month-old pineapple plants were exposed to various stress treatments, and the leaves were collected for qRT-PCR analysis. After low temperature treatment, the expression of *AcWRKY31* showed the minimum level at 2 h, then increased gradually, and reached the maximum level at 12 h (Figure 3A). High temperature treatment could inhibit the expression level of *AcWRKY31*, and *AcWRKY31* reached its minimum expression level at 48 h (Figure 3B). Under drought treatment, the expression level of *AcWRKY31* reached the maximum level at 6 h, then declined gradually, and reached the minimum level at 12 h (Figure 3C). Under salt treatment, the expression level of *AcWRKY31* was inhibited at 2 and 6 h and then increased significantly at 12 h (Figure 3D). These results suggest that AcWRKY31 may be involved in abiotic stress response.

### 2.4. Phenotype Observation of AcWRKY31 Overexpression Rice Transgenic Plant

Since both rice and pineapple are important monocotyledonous plants, in order to further investigate the function of *AcWRKY31*, we constructed *AcWRKY31* overexpression rice transgenic plants and obtained two independent lines, OE-3 and OE-31 (Appendix A). It was found that the panicle length of OE-3 and OE-31 was shorter than that of ZH11 (Appendix A). In addition, the statistical results showed that the total seed number of OE-3 and OE-31 was significantly decreased compared with that in ZH11 (Appendix A). These results indicate that overexpression of *AcWRKY31* in rice may affect plant growth and development. 

### 2.5. Analysis of Differentially Expressed Genes (DEGs) in AcWRKY31 Rice Transgenic Plants

To investigate the mechanism of *AcWRKY31* regulating rice growth and development, total RNA-seq analysis was performed to analyze the transcriptome in the leaves of ZH11 and *AcWRKY31* overexpression transgenic rice plants. Differential gene expression analysis showed that 1441 genes were upregulated and 1126 genes were downregulated in transgenic rice plants compared with the corresponding gene in ZH11 (Appendix A), and their putative functions are shown in Appendix A. To verify the accuracy of the RNA-seq data, we performed qRT-PCR to compare the expression levels of six differential expressed genes, including *LOC_Os01g50890*, *LOC_Os10g40700*, *LOC_Os11g35300*, *LOC_Os01g06310*, *LOC_Os11g05470* and *LOC_Os11g06150*. The expression trend of these six selected genes was consistent with the results of RNA-seq, which indicated the reliability of the RNA-seq data and its subsequent analysis (Appendix A).

Gene ontology (GO) enrichment analysis described the biological process, cellular component and molecular function of these 1457 upregulated genes. Among the biological processes, these DEGs were mainly concentrated in the carbohydrate metabolism process, response to endogenous stimulus, and response to stimulus. The cell components of these DEGs were mainly distributed in an external encapsulating structure, cell wall, and extracellular region. The molecular function of DEGs included hydrolase activity, catalytic activity and oxygen binding (Figure 4A). Kyoto Encyclopedia of Genes and Genomes (KEGG) pathway analysis indicated that these upregulated genes were mainly distributed in phenylpropanoid biosynthesis, plant hormone signal transduction, and starch and sucrose metabolism (Figure 4B). The GO analysis showed that 1140 downregulated genes were mainly concentrated in the 21 GO terms, including secondary metabolic process, response to abiotic stimulus, mitochondrion and transferase activity (Figure 4C). Furthermore, the KEGG analysis showed that the downregulated genes were assigned to eight KEGG pathways, including glutathione metabolism, biotin metabolism and MAPK signaling pathway (Figure 4D).

### 2.6. Overexpression of AcWRKY31 Decreased Drought and Salt Tolerance in Transgenic Rice Plants 

To further confirm whether *AcWRKY31* is involved in the process of drought stress response, the seedings of ZH11, OE-3 and OE-31 were seeded in 1/2 MS medium with or without 200 mM mannitol. The results showed that the growth of *AcWRKY31* overexpression rice seedlings was found to be severely repressed by application of 200 mM mannitol compared to control plants (Figure 5). OE-3 and OE-31 exhibited lower plant height, root length and fresh weight compared with those in wild-type plant ZH11 under drought stress condition, which indicated that overexpression of *AcWRKY31* increases rice sensitivity to drought mimicked by mannitol application (Figure 5).

To explore whether *AcWRKY31* is also involved in the process of salt stress response, the seedings of ZH11, OE-3 and OE-31 were seeded in 1/2 MS medium with or without 150 mM NaCl. The results showed that the plant height, root length and fresh weight of *AcWRKY31* overexpression rice seedlings OE-3 and OE-31 was significantly reduced compared with those in ZH11 after salt stress treatment, which indicated that overexpression of *AcWRKY31* increases rice sensitivity to salt stress (Figure 6).

### 2.7. Overexpression of AcWRKY31 Decreased Drought and Salt Tolerance in Transgenic Arabidopsis Plants 

In order to investigate whether heterologous overexpression of *AcWRKY31* in *Arabidopsis* also has similar function, we generated two *AcWRKY31* transgenic *Arabidopsis* lines, OE-10 and OE-12 (Appendix A). The seedings of WT, OE-10 and OE-12 were planted in the 1/2 MS medium as the control group. Volumes of 200 and 250 mM mannitol were used to simulate drought treatment. Then, 1/2 MS medium containing 75 mM and 100 mM NaCl was used to mimic salt treatment. The results showed that compared with the WT, the germination rate, root length and fresh weight of OE-10 and OE-12 were decreased under 200 mM mannitol and 75 mM NaCl treatments, and further decreased under 250 mM mannitol and 100 mM NaCl treatments (Figure 7 and Figure 8).

### 2.8. Expression Analysis of Stress-Related Genes in AcWRKY31 Transgenic Plants 

To investigate the potential molecular pathway affected by *AcWRKY31* in regulating stress tolerance, two-week-old ZH11, OE-3 and OE-31 seedings were exposed to drought and salt treatments, respectively. We performed qRT-PCR to monitor the expression levels of six abiotic stress-responsive genes in rice plants, including *OsAPX2*, *OsCAT1*, *OsCATB*, *OsDREB2A*, *OsDREB2B* and *OsABA1*. These results showed that compared with ZH11, the expression levels of *OsAPX2*, *OsCAT1*, *OsCATB*, *OsDREB2A*, *OsDREB2B* and *OsABA1* were decreased in OE-3 and OE-31 under normal conditions (Figure 9). Furthermore, after drought and salt treatments, the expression levels of these abiotic stress-responsive genes were increased in OE-3 and OE-31, and further increased in ZH11 (Figure 9).

Furthermore, we also performed qRT-PCR to analyze the expression levels of five stress-related genes (*AtCAT1*, *AtCAT3*, *AtPOD1*, *AtPOD2* and *AtRD22*) in WT, OE-10 and OE-12. It was found that the expression levels of *AtCAT1*, *AtCAT3*, *AtPOD1*, *AtPOD2* and *AtRD2* were inhibited in OE-10 and OE-12 compared with those in WT under normal condition (Figure 10). Furthermore, the expression levels of these genes were increased in the OE-10 and OE-12 and more strongly increased in WT after drought and salt treatments (Figure 10). These results showed that the expression levels of the stress-related genes were affected upon *AcWRKY31* overexpression in rice and *Arabidopsis*.

## 3. Discussion

Plant growth, yield and quality are seriously threatened by various abiotic stresses [25,26]. To cope with these environmental challenges, plants have to develop effective stress response mechanisms [27,28]. As one of the TF superfamilies in plants, WRKY TFs are involved in various plant biological processes [2,3,4]. At present, a large number of studies have reported that WRKY TFs play important roles in various plant species growth and development, such as soybean (*Glycine max*), rice (*Oryza sativa* L.) and cotton (*Gossypium barbadense* L.) [29,30,31,32]. Pineapple is an important tropical and subtropical fruit with high nutritional and economic value. It is widely planted in China and other countries. However, there are few studies on the function of the *WRKY* gene in pineapple. In this study, *AcWRKY31* was isolated and cloned from the pineapple genome. The sequence alignment of AcWRKY31 exhibited that the AcWRKY31 protein contained a highly conserved WRKY domain and C2HC-type of zinc-finger motif. AcWRKY31 shared high sequence similarity with AcWRKY25, OsWRKY113 and AtWRKY53 in pineapple, rice and *Arabidopsis*, respectively (Figure 1). Several studies have reported that *AcWRKY25*, *OsWRKY113* and *AtWRKY53* are involved in stress response. The expression of *AcWRKY25* was induced by cold stress [20]. *OsWRKY113* was involved in iron toxicity tolerance mechanisms [33]. *AtWRKY53* plays an important role in the senescence of plants, and activated expression of *AtWRKY53* negatively regulates plant drought tolerance [13,34]. In the present study, the expression profiles of the pineapple *AcWRKY31* gene show some kind of “up and down” patterns after different stress treatments (Figure 3), which is similar to the previous study [20]. The change in *AcWRKY31* expression may be caused by the plant circadian clock, which can coordinate the internal metabolic and physiological processes of plant and change gene expression to adapt to the external environment [35].

Increasing evidence have demonstrated that WRKY TFs play important roles in plant growth and development. It has been demonstrated that *AtWRKY71*, *GsWRKY20* and *OsWRKY11* are crucial components affecting plant flowering [36,37,38]. The rice gene *OsWRKY78* was involved in seed germination and stem elongation [30]. Overexpression of *TaWRKY71* in wheat can improve the rate of seed germination [39]. *AtWRKY75* can not only regulate the aging process of plant leaves, it can also affect the elongation of plant roots [40]. Since the genetic transformation system of pineapple is still limited, we constructed *AcWRKY31* overexpression transgenic plants to further study the function of *AcWRKY31*. In this study, we found that overexpression of *AcWRKY31* in rice will affect plant panicle length and total seed number (Appendix A). Furthermore, *AcWRKY31* was also involved in plant stress response. Under drought and salt conditions, the plant height, root length and fresh weight of *AcWRKY31* overexpression rice plants were reduced significantly compare with those in ZH11 (Figure 5 and Figure 6). After drought and salt stress treatment, *AcWRKY31* overexpression *Arabidopsis* plants exhibited lower seed germination rate, shorter root length and lighter fresh weight compared with those in WT (Figure 7 and Figure 8). These results showed that the transgenic plants with higher *AcWRKY31* expression levels were more sensitive to drought and salt stress, which indicated that overexpression of *AcWRKY31* inhibited plant growth and development under drought and salt stresses.

When plants are exposed to unfavorable environmental conditions, they can establish an effective defense mechanism depending on the precise regulation of various stress-responsive genes, including hormones biosynthesis, signaling transduction and osmoprotectants metabolism-related genes, and this regulatory mechanism comprises a series of transcriptional activators or repressors [41,42]. The WRKY transcription factors have been well recognized for their roles in a regulatory network that integrates internal and environmental factors to regulate plant stress tolerance [43,44]. For instance, overexpression of *TaWTKY19* in *Arabidopsis* enhanced plant salt and drought tolerance by upregulating the expression level of *DREB2A*, *RD29A* and *RD29B* [45]. Overexpression of *GsWRKY20* and *GmWRKY16* in *Arabidopsis* can enhance the drought tolerance of transgenic *Arabidopsis* plants through the ABA signaling pathway [46,47]. Conversely, ectopic overexpression of *GhWRKY33* can enhance the sensitivity of transgenic *Arabidopsis* to drought stress by downregulating the expression of several stress-responsive genes, such as *RD29A*, *DREB2A* and *ABI1* [48]. Similarly, we performed qRT-PCR to investigate the expression level of several stress-related genes in *AcWRKY31* transgenic plants and wild-type plant controls. In our study, the expression levels of various stress-responsive genes, including *OsAPX2*, *OsCAT1*, *OsCATB*, *OsDREB2A*, *OsDREB2B* and *OsABA1* in rice, *AtCAT1*, *AtCAT3*, *AtPOD1*, *AtPOD2* and *AtRD22* in *Arabidopsis*, were significantly lower in *AcWRKY31* transgenic plants than those in wild-type plants under normal or stress conditions (Figure 9 and Figure 10). These findings indicated that *AcWRKY31* affects plant stress tolerance by altering the expression of the stress-related genes.

Taken together, the above findings revealed the potential function of pineapple *AcWRKY31*. Compared with wild-type plants, *AcWRKY31* overexpression plants are more sensitive to drought and salt stresses, which may be related to the decrease in the expression levels of stress-related genes. This study provided a theoretical foundation for the further functional characterization of *AcWRKY31*. However, much more work needs to be conducted to further investigate the molecular mechanisms of *AcWRKY31* under stress conditions. 

## 4. Materials and Methods

### 4.1. Plant Materials and Abiotic Treatments of Pineapple

*Arabidopsis thaliana* ecotype Columbia-0 (Col-0) and rice (*Oryza sativa* L.) ZH11 (Zhong Hua 11) were used as wild-type plants in this study. *AcWRKY31* transgenic rice and *Arabidopsis* lines were obtained through *Agrobacterium*-mediated transformation [49,50]. The T3 generation transgenic lines with high transcription level were obtained for further analysis. Pineapple (*A*. *comosus* var MD-2) was provided by the Qin Lab (Center for Genomics and Biotechnology, Fujian Agriculture and Forestry University, Fujian, China). The pineapple seedings were grown under 16 h light/8 h dark photoperiod and 70% relative humidity at 25 ℃. The one-month-old pineapple plants were exposed to the following treatments: cold stress (4 ℃), heat stress (45 ℃), drought stress (350 mM mannitol) and salt stress (150 mM NaCl). The pineapple leaves were collected after 2, 6, 12, 24 and 48 h treatments and were stored at −80 ℃ for subsequent analysis.

### 4.2. Bioinformatics Analysis

The AcWRKY31 protein and its homologous protein sequences were downloaded from Phytozome13 (https://phytozome-next.jgi.doe.gov/) (accessed on 25 May 2022). Multiple sequence alignment was performed by DNAMAN software (version 9) [51].

### 4.3. Vector Construction and Subcellular Localization

The coding sequences of *AcWRKY31* gene was amplified from pineapple cDNA using the primers listed in Appendix A. The PCR fragment was constructed into pENTR™/D-TOPO vector (CAT: K2400-20, Invitrogen), and then recombined into the plant expression vector pGWB605 with CaMV 35S promoter and GFP (green fluorescent protein) using LR clone II enzyme (Invitrogen). The vector *35S::AcWRKY31::GFP* was transformed into the *Agrobacterium tumefaciens* GV3101 and then infiltrated to *Nicotiana benthamiana* leaves with the infection buffer (10 mM MES, 50 mM MgCl2, pH = 5.8, 100 µm AcetoSyringone) [52,53,54]. The *35S::GFP* empty vector was used as a negative control. After dark growth for 36 to 48 h, the fluorescence signal of the recombinant proteins in the leaves was observed using LAICA SP8 confocal microscope, with a 488 nm wavelength for GFP signal.

### 4.4. Transactivation Activity Assays 

The CDS of *AcWRKY31* was introduced into the pGBKT7 vector to generate pGBKT7-AcWRKY31. The yeast strain AH109 was transformed with pGBKT7, pGBKT7-AcWRKY31 and pGBKT7-AcWRKY31 + pGADT7-T. The transformed cells were grown on SD/-Trp/-Leu for 2 to 4 days and then transferred to SD/-Trp/-Leu/-His/X-α-gal medium for further culture. The transcriptional activity of proteins was detected by the yeast strains’ growth status and X-α-gal activity.

### 4.5. Assessment of Drought and Salt Tolerance in Transgenic Plants

Seeds of ZH11 and *AcWRKY31* overexpression transgenic rice plants were planted in the sterilized glass cans containing 1/2 Murashige and Skoog (MS) medium with 200 mM mannitol and 150 mM NaCl, respectively. The control group did not receive any stress treatment. The seedings grew under 16 h light/8 h dark photoperiod at 30 ℃. Plant height, root length and fresh weight were measured after 10 days. For *Arabidopsis*, WT and the transgenic lines were planted in the 1/2 MS medium as the control group. Volumes of 200 and 250 mM mannitol were used to simulate drought treatment. Then, 1/2 MS medium containing 75 and 100 mM NaCl were used to mimic salt treatment. The seedings grew under a 16 h light/8 h dark photoperiod at 22 ℃. After 7 days, plant fresh weight and germination rates were counted, and the root length was measured.

For the analysis of stress-related gene expression, the two-week-old rice seedings were grown in the nutrient solution with or without 200 mM mannitol or 150 mM NaCl [55]. For *Arabidopsis*, one-month-old WT and transgenic plants were grown in soil watered with deionized water with or without 200 mM mannitol or 150 mM NaCl. The leaves of rice and *Arabidopsis* were collected at 0 and 6 h post-treatment for further analysis.

### 4.6. RNA-Seq and Data Analysis 

RNA was isolated from the leaves of ZH11 and *AcWRKY31* overexpression (*AcWRKY31-OE*) rice plants using plant RNA extraction kit (OMEGA, Shanghai, China) following the manufacturer’s protocol. The RNA-seq data of rice leaves were downloaded from Phytozome13, and the sequencing and data processing were conducted as previously described [56]. We used the TRIMMOMATIC v0.3 to filter the raw reads and remove the adapter sequence [57]. The clean reads were aligned using Tophat software with default parameters, and then the transcripts were assembled and quantified using Cufflinks [58]. Differentially expressed genes (DEGs) were obtained by using Cuffdiff (fold change ≥ 2; a value of FDR ≤ 0.05 was considered to be statistically significant) [58]. Gene ontology (GO) and Kyoto Encyclopedia of Genes and Genomes (KEGG) analysis of DEGs was performed using TBtools v1.09 software [59]. The final results were visualized by using R package UpSet v1.0.0.

### 4.7. Quantitative Real-Time PCR Analysis 

Total RNA was isolated using TRIzol (Invitrogen, Carlsbad, CA, USA) and were reverse-transcribed using AMV reverse transcriptase (Takara, Japan), following the manufacturer’ instructions [53]. Quantitative real-time PCR was performed based on the SYBR Premix Ex Taq II system (Takara, Japan) and Bio-Rad Real-Time PCR system. The reaction was carried out in a 20 µL volume containing 10 µL of 2× SYBR Premix, 8.2 µL of RNase free water, 1 µL of template, 0.4 µL of each specific primer (Appendix A), and performed with the following parameters: 95 ℃ for 30 s; 40 cycles of 95 ℃ for 5 s and 60 ℃ for 34 s; 95 ℃ for 15 s [60,61]. The genes *AcPP2A*, *AtHK2* and *OsUBQ5* were used as reference genes in pineapple, *Arabidopsis* and rice, respectively [54,62,63]. Three biological replicates were performed, and every biological replicate was confirmed by three technical replicates. The relative expression levels of these selected genes were calculated using the comparison threshold period (2^−∆∆Ct^) method [64].

## 5. Conclusions

In this study, we cloned and characterized pineapple *AcWRKY31*. Our results revealed that ectopic overexpression of *AcWRKY31* will reduce plant drought and salt tolerance by altering the expression levels of stress-responsive genes in transgenic rice and *Arabidopsis*. These findings enhanced the understanding of the role of pineapple AcWRKY31 transcription factor in the complex abiotic stress molecular mechanisms, and provided a theoretical basis for the functional characterization of *AcWRKY31* genes in different plant species.

## Figures and Tables

**Figure 1 ijms-23-06269-f001:**
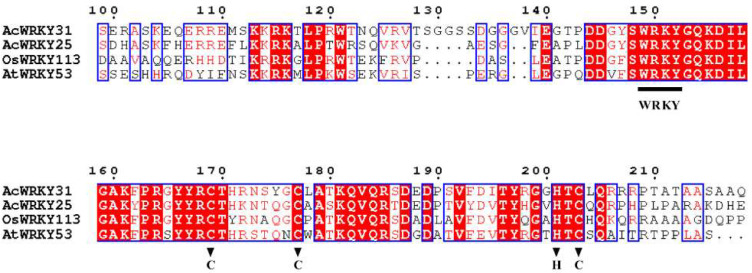
Multiple sequence alignments of AcWRKY31 and its ortholog proteins. The WRKY domain is indicated by the black line, and black triangles indicate the C2HC zinc-finger motif.

**Figure 2 ijms-23-06269-f002:**
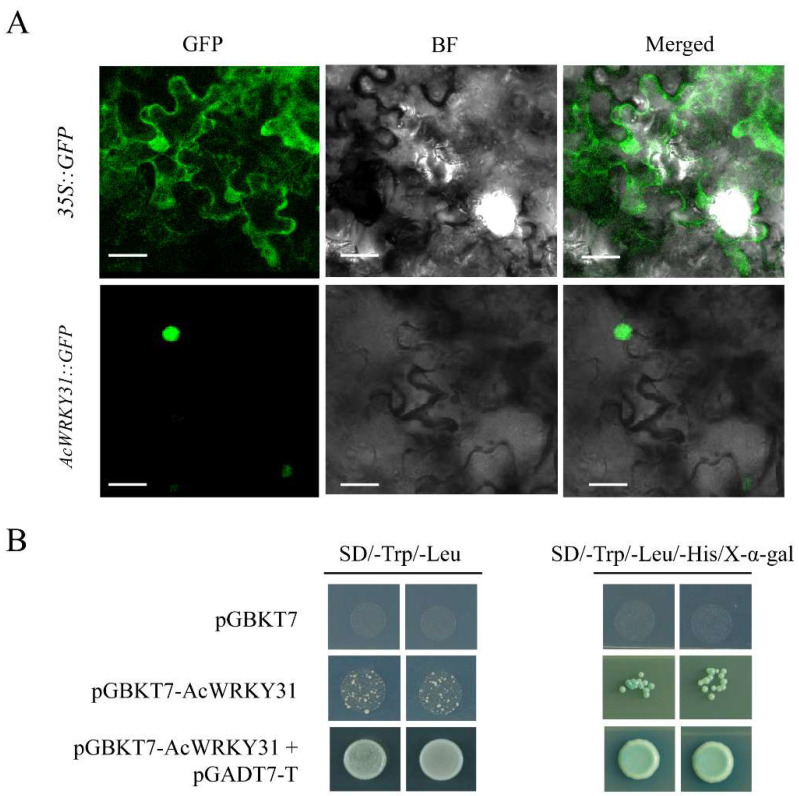
Subcellular localization and transcriptional activation analysis of AcWRKY31. (**A**) The fusion proteins *35S::*GFP and AcWRKY31::GFP were transiently expressed in in *Nicotiana benthamiana* leaves cells and observed with a laser scanning confocal microscope. Scale bar = 50 μm. (**B**) Transcriptional activation analysis of AcWRKY31.

**Figure 3 ijms-23-06269-f003:**
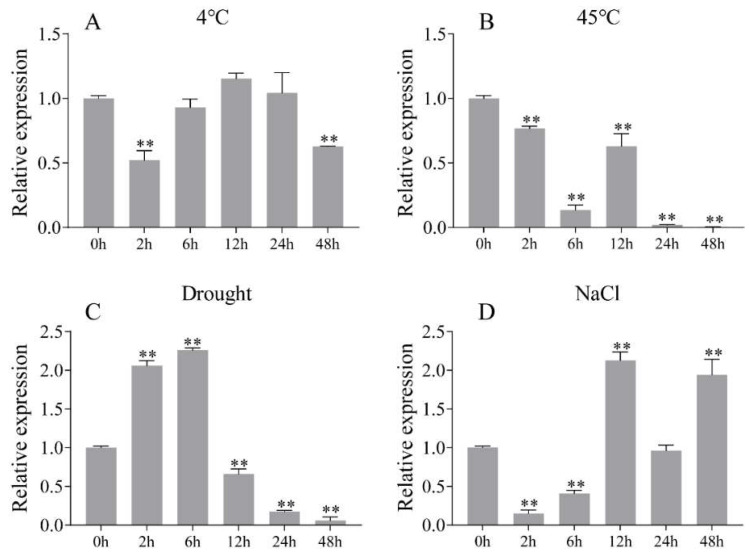
The expression pattern analysis of *AcWRKY31* under different stress treatments. The one-month pineapple seedings were treated with cold (**A**), heat (**B**), drought (**C**) and salt stress (**D**). The error bars indicate ± SD (*n* = 3). Asterisks indicate significant differences for the indicated comparisons based on Student’s *t* test (** *p* < 0.01).

**Figure 4 ijms-23-06269-f004:**
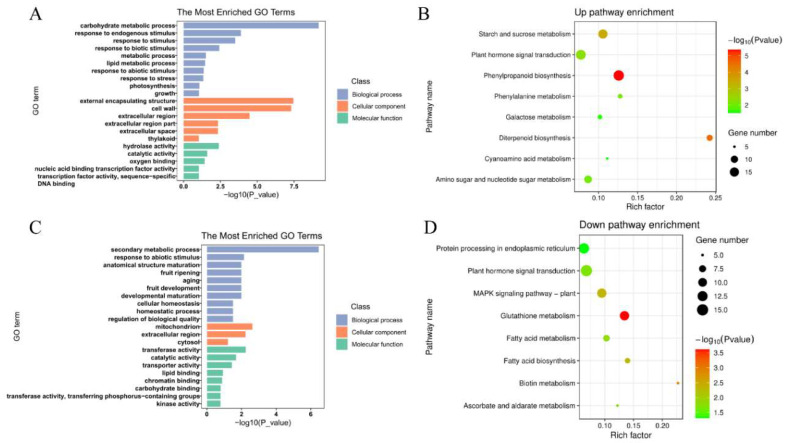
Transcriptomic analysis of ZH11and *AcWRKY31-OE* transgenic rice plants. (**A**) GO analysis of the upregulated genes. (**B**) KEGG pathway analysis for upregulated genes. (**C**) GO analysis of the downregulated genes. (**D**) KEGG pathway analysis for downregulated genes.

**Figure 5 ijms-23-06269-f005:**
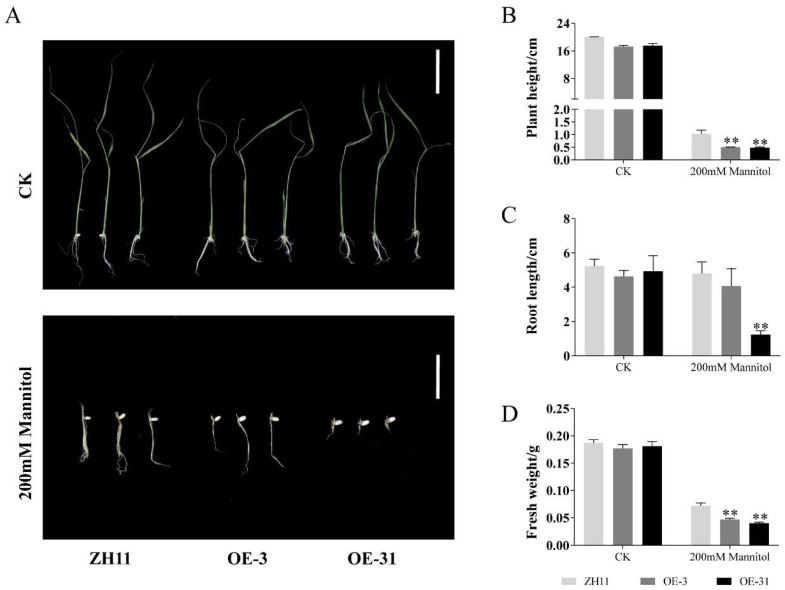
Drought stress tolerance of *AcWRKY31* overexpression rice plants. (**A**) Performance of ZH11 and *AcWRKY31* overexpression rice plants with or without drought treatment for 10 days; scale bar = 5 cm. The plant height (**B**), root length (**C**) and fresh weight (**D**) of all lines in (**A**). Bars show standard deviations of at least 10 seedlings. Asterisks indicate significant differences between the ZH11 and the *AcWRKY31* overexpression lines evaluated with Student’s *t* test (** *p* < 0.01).

**Figure 6 ijms-23-06269-f006:**
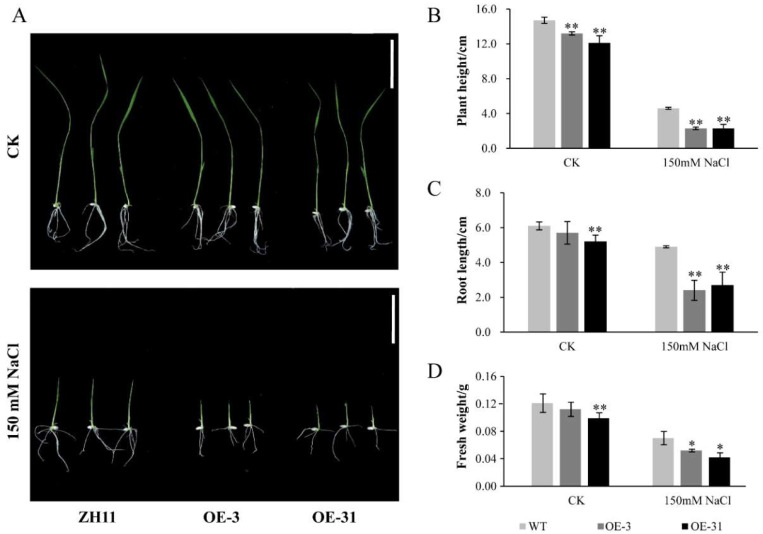
Salt stress tolerance of *AcWRKY31* overexpression rice plants. (**A**) Performance of ZH11 and *AcWRKY31* overexpression rice plants with or without salt treatment for 10 days; scale bar = 5 cm. The plant height (**B**), root length (**C**) and fresh weight (**D**) of all lines in (**A**). Bars show standard deviations of at least 10 seedlings. Asterisks indicate significant differences between the ZH11 and the *AcWRKY31* overexpression lines evaluated with Student’s *t* test (* *p* < 0.05, ** *p* < 0.01).

**Figure 7 ijms-23-06269-f007:**
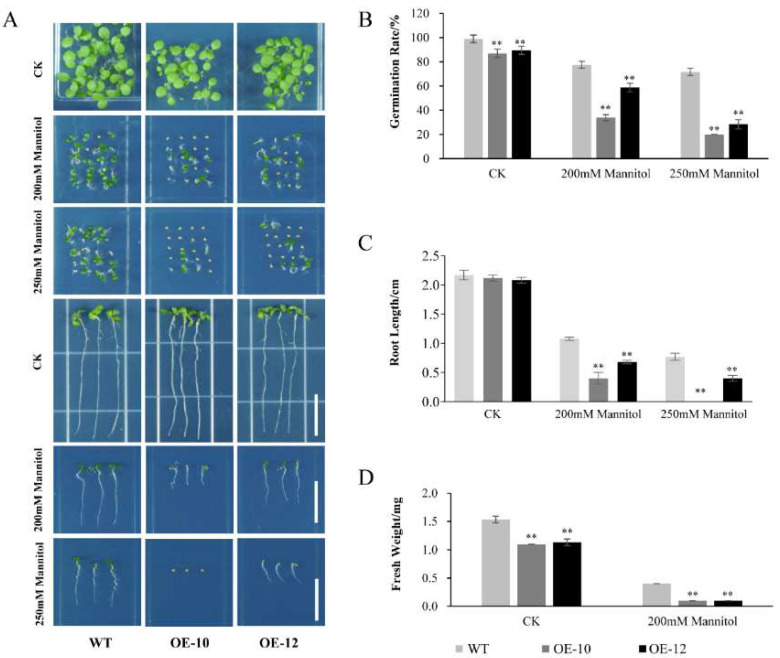
Phenotypic analysis of *AcWRKY31* overexpression *Arabidopsis* plants in response to drought stress. (**A**) Performance of WT and *AcWRKY31* overexpression *Arabidopsis* plants with or without drought treatment for 7 days; scale bar = 1 cm. The germination rate (**B**), root length (**C**) and fresh weight (**D**) of all lines in (**A**). Error bars indicate ± SD of three biological replicates. Asterisks indicate significant differences between the WT and the *AcWRKY31* overexpression lines evaluated with Student’s *t* test (** *p* < 0.01).

**Figure 8 ijms-23-06269-f008:**
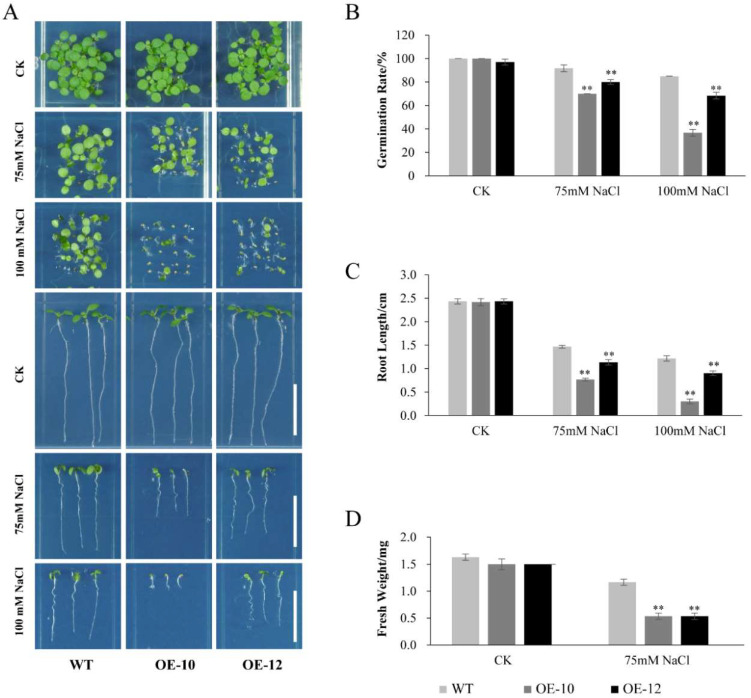
Phenotypic analysis of *AcWRKY31* overexpression *Arabidopsis* plants in response to salt stress. (**A**) Performance of WT and *AcWRKY31* overexpression *Arabidopsis* plants with or without salt treatment for 7 days; scale bar = 1 cm. The germination rate (**B**), root length (**C**) and fresh weight (**D**) of all lines in (**A**). Error bars indicate ± SD of three biological replicates. Asterisks indicate significant differences between the WT and the *AcWRKY31* overexpression lines evaluated with Student’s *t* test (** *p* < 0.01).

**Figure 9 ijms-23-06269-f009:**
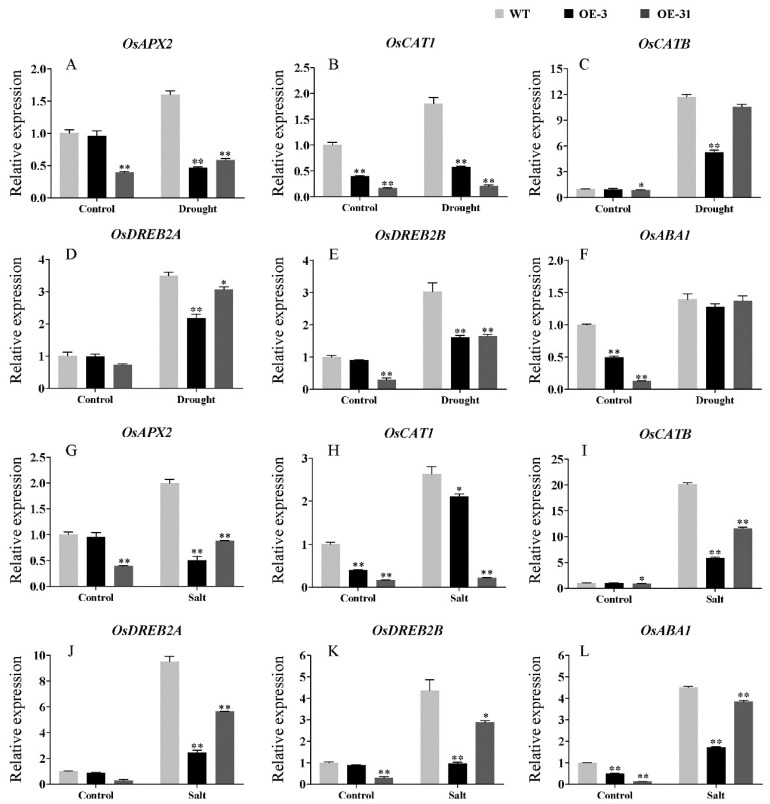
The expression of abiotic stress-related genes in the ZH11 and *AcWRKY31* transgenic rice plants in response to drought (**A**–**F**) and salt stresses (**G**–**L**). The error bars indicate ± SD (*n* = 3 replicates). Asterisks indicate significant differences for the indicated comparisons based on Student’s *t* test (* *p* < 0.05, ** *p* < 0.01).

**Figure 10 ijms-23-06269-f010:**
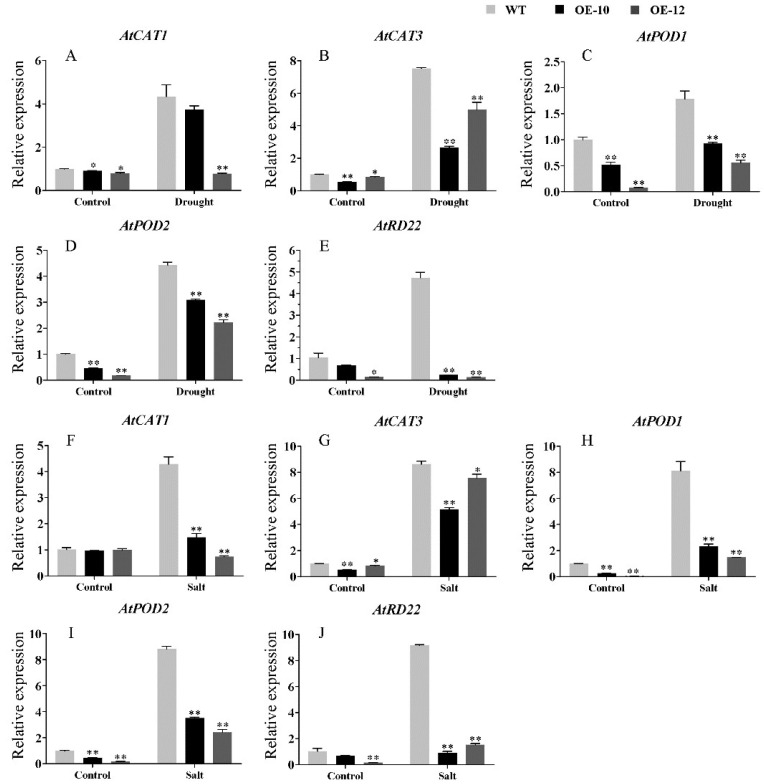
The expression of abiotic stress-related genes in the WT and *AcWRKY31* transgenic *Arabidopsis* plants in response to drought (**A**–**E**) and salt stresses (**F**–**J**). The error bars indicate ± SD (*n* = 3 replicates). Asterisks indicate significant differences for the indicated comparisons based on Student’s *t* test (* *p* < 0.05, ** *p* < 0.01).

## Data Availability

All data analyzed during this study are included in this article and its additional files.

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
