# Peer review of "Ectopic Overexpression of Pineapple Transcription Factor AcWRKY31 Reduces Drought and Salt Tolerance in Rice and Arabidopsis"

_ijms, 2022, doi:10.3390/ijms23116269_

Round 1

Reviewer 1 Report

The reviewed manuscript deals with a very interesting issue, which is environmental stress. Of course, there are many factors that cause stress, but you have chosen very important from your point of view. Indeed, drought and salt stress in warm conditions pose serious problems. Your work is prepared very carefully, the descriptions of your methods that you used deserve attention. Your figures deserve attention, as they perfectly illustrate your research and indicate how important the issue is the molecular study of genes responsible for environmental stress. I will recommend the editorial board to accept your work for publication without changes.

Author Response

Response to Reviewer 1 Comments:

Point 1: The reviewed manuscript deals with a very interesting issue, which is environmental stress. Of course, there are many factors that cause stress, but you have chosen very important from your point of view. Indeed, drought and salt stress in warm conditions pose serious problems. Your work is prepared very carefully, the descriptions of your methods that you used deserve attention. Your figures deserve attention, as they perfectly illustrate your research and indicate how important the issue is the molecular study of genes responsible for environmental stress. I will recommend the editorial board to accept your work for publication without changes.

Response 1: We sincerely appreciate for your time and attention you have spent on this manuscript, and thank you for your high evaluation of our research.

Reviewer 2 Report

Hereunder are my remarks that, in my opinion, can improve the manuscript:

I think it is easier and more practical for the reader to see the figures in the results section. Thus, I recommend for authors to add figures in the body of the text.

Although you presented briefly the main findings of your work at the end of the introduction, please emphasize the originality of your work, by using for instance the following beggining of sentence; "This is the first work showing/reporting…."

Line 94: I think it could be better to replace ‘it is reported’ by ‘several studies/previous studies have reported’.

Line 131: Please remove the point after “response” and replace it by a comma. Thus, the comma after ZH11 needs to be removed so.

Line 142: “also have” should become “also has”.

Line 157: “to analysis” should become “to analyze”

Line 193:  Please remove the “s” of “enhances”, so “can enhances” becomes “can enhance”.

Material and methods section need reorganization. The plant material section should appear first before all sections. Then, the order of appearance of other sections should be the same than the one of results.

Line “257”: Please remove “from the” and “by”. In this line, “reversed-transcribed” should become “reverse-transcribed” (without the ‘d’).

In the results section, the observation and interpretation of Figures 1 B and C needs to be more developed. Authors were succinct describing the results of the subcellular localization and transcriptional activity.

Photos of Figure 1C NEED be changed ! The quality is low, They are not clear AT ALL!

This work contains a respectful amount of data. However,  I think that the discussion of the results was so brief. Authors should go in details in the when discussing results.

In supplementary figures 1 and 2, authors need to add the type of statistical test realized, the signification of the two stars, the p values (p<..) AND the standard deviation (positive and negative parts).

Good Luck,

Author Response

Response to Reviewer 2 Comments:

Hereunder are my remarks that, in my opinion, can improve the manuscript:

Point 1: I think it is easier and more practical for the reader to see the figures in the results section. Thus, I recommend for authors to add figures in the body of the text.

Response 1: Thank you for your suggestion, we have added the figures in the body of the text.

Point 2: Although you presented briefly the main findings of your work at the end of the introduction, please emphasize the originality of your work, by using for instance the following beggining of sentence; "This is the first work showing/reporting…."

Response 2: Thanks for your good advice. We re-emphasize the originality of our work in the introduction section (Line 125-128).

Point 3: Line 94: I think it could be better to replace ‘it is reported’ by ‘several studies/previous studies have reported’.

Response 3: Thank you for you suggestion, we have revised this sentence according to your advice (Line 346).

Point 4:  Line 131: Please remove the point after “response” and replace it by a comma. Thus, the comma after ZH11 needs to be removed so.

Response 4: Thank you for your suggestion, we have revised this sentence according to your advice (Line 412).

Point 5: Line 142: “also have” should become “also has”.

Response 5: Thank you for your suggestion, we have revised this sentence according to your advice (Line 457).

Point 6:  Line 157: “to analysis” should become “to analyze”

Response 6: We are very sorry for the spelling mistakes in the manuscript, we have revised this sentence according to your advice (Line 502).

Point 7:  Line 193:  Please remove the “s” of “enhances”, so “can enhances” becomes “can enhance”.

Response 7: We are very sorry for this mistake in the manuscript, we have revised this sentence according to your advice (Line 305).

Point 8:  Material and methods section need reorganization. The plant material section should appear first before all sections. Then, the order of appearance of other sections should be the same than the one of results.

Response 8: Thanks for your suggestion. In the revised version, we have carefully revised the section of “Materials and Methods” to ensure that the order of the “Materials and Methods” corresponds to the “Result” section.

Point 9:  Line “257”: Please remove “from the” and “by”. In this line, “reversed-transcribed” should become “reverse-transcribed” (without the ‘d’).

Response 9: We apologize for this mistake in the manuscript. We have revised this word in the revised version (Line 755).

Point 10:  In the results section, the observation and interpretation of Figures 1 B and C needs to be more developed. Authors were succinct describing the results of the subcellular localization and transcriptional activity.

Response 10: Thank you for your professional comments. Based on this comment, we have added more detail to describe the results of the subcellular localization and transcriptional activity of AcWRKY31 in the result section.

Point 11: Photos of Figure 1C NEED be changed ! The quality is low, They are not clear AT ALL!

Response 11: We are very sorry that the resolution of “Figure 1C” is low in original manuscript. We have revised it in the resubmitted manuscript (Figure 2B). We are sincerely hope the revised manuscript have met the review requirements.

Point 12: This work contains a respectful amount of data. However, I think that the discussion of the results was so brief. Authors should go in details in the when discussing results.

Response 12: Thanks for your suggestion. We have added more detail to describe the results in the discussion section.

Point 13: In supplementary figures 1 and 2, authors need to add the type of statistical test realized, the signification of the two stars, the p values (p<..) AND the standard deviation (positive and negative parts).

Response 13: Thanks for your suggestion, we have added the statistical test method, the standard deviation and the significant differences of supplementary figures 1-3 in the revised version (Line 866-881).

Reviewer 3 Report

This paper is focused on one WRKY transcription factor from pineapple. A major issue is the protein identification. On line 78, it is stated that the protein is AcWRKY50 (Aco005719.1 gene). If one refers to the Xie et al. paper (Genome-wide investigation of WRKY gene family in pineapple: evolution and expression profiles during development and stress. BMC Genomics 19, 1-18), the Aco005719.1 gene encodes AcWRKY28, not AcWRKY50 (Xie's paper additional file 1). This inversion is confirmed by the partial amino acid sequence given in figure 1, which is indeed the AcWRKY28 sequence. This obviously requires a throughout edition of the article.

The pineapple WRKY TF family has 54 members. The authors do not justify the choice of the gene they have decided to study. If one looks at the Xie et al. paper again (fig. 7c), the AcWRKY28 and AcWRKY50 genes are reported to be the most induced by cold. Whatever the motivation behind the choice of the gene, it has to be stated.

The nuclear localisation of AcWRKY28 in pineapple cells has already been demonstrated in the following paper by some of the co-authors: Priyadarshani et al. (2019) Biomolecules 9, 617. Therefore, this paper should be cited and the nuclear localisation experiment in tobacco is no longer relevant and should be deleted.

To demonstrate the presence of a transcription activation domain in the WRKY28 protein, the authors fused its coding sequence to the yeast Gal DNA-binding domain coding sequence of pGBKT7. It then makes no sense to transform yeast with both this plasmid and pGADT7-T which contains the Gal activation domain. Only pGBKT7-AcWRKY28 and pGBKT7 as a control should have been used in this experiment.

The expression profiles of the pineapple AcWRKY28 gene after stresses show some kind of "up and down" patterns that are very reminiscent of the diurnal variations described in the following paper (that should be cited): Sherma et al. (2017) Diurnal cycling transcription factors of pineapple revealed by genome-wide annotation and global transcriptomic analysis. Genome Biol. Evol. 9:2170–2190. According to this work, AcWRKY28 shows a diurnal cycling of expression. The lack of induction of expression of AcWRKY28 by cold should be discussed in relation with the data published by Xie et al. (fig 7c).

There is little information about the rice and Arabidopsis AcWRKY28-overexpressing lines used here: which promoters were used to drive the expression? Were the levels of expression different between the two lines used per species? Were the AcWRKY28 level of expression correlated with the level of expression of some of the effector genes? The details of the constructs and the transgene expression data should be added.

From this work, it is difficult to draw any conclusion about the function of the AcWRKY28 gene in pineapple. Yes, its overexpression reduced the tolerance of heterologous transgenic plants to various stresses but there could be several possible explanations for this. Some WRKY TF are known to act as negative feedback regulators of MPK (Chen et al. (2019) WRKY transcription factors: evolution, binding, and action. Phytopathology Research 1:13). The TF genes are generally expressed at low level. The abnormal abundance of a TF in overexpressing plants could result in the disruption of pathways involving other endogenous TF. These disruptions could explain the decrease of stress resistance in overexpressing plants although this mode of action might not be relevant in plants producing the TF at a normal level. Similarly, some WRKY TF are known to interact with kinases. One can easily imagine how an abnormal abundance of a TF in overexpressing plants could disrupt endogenous pathways by acting as a "trap" for TF-interacting kinases. Therefore, the conclusion (line 183-184) "These results suggest that AcWRKY50 serves as a negative regulator of plant growth and development under drought and salt stresses" should be mitigated because there is no direct evidence for this from the data. The evidence could only be brought up by the study of an AcWRKY28 KO pineapple plant.

The supplementary table 2 is of little use without any gene function mentioned. There should be an extra column indicating the putative function of each gene.

The quality of the writing is generally good but a few sentences need editing. Here are two examples from the abstract:

- line 19: ...overexpression transgenic rice plants WERE significantly reduced

- line 21-22: ...ectopic overexpression of AcWRKY50 in rice and Arabidopsis exhibited RESULTED IN PLANT OVERsensitivity to drought and salt stressES

Author Response

Response to Reviewer 3 Comments:

Point 1: This paper is focused on one WRKY transcription factor from pineapple. A major issue is the protein identification. On line 78, it is stated that the protein is AcWRKY50 (Aco005719.1 gene). If one refers to the Xie et al. paper (Genome-wide investigation of WRKY gene family in pineapple: evolution and expression profiles during development and stress. BMC Genomics 19, 1-18), the Aco005719.1 gene encodes AcWRKY28, not AcWRKY50 (Xie's paper additional file 1). This inversion is confirmed by the partial amino acid sequence given in figure 1, which is indeed the AcWRKY28 sequence. This obviously requires a throughout edition of the article.

Response 1: We sincerely appreciate for your suggestion. Since our laboratory is also conducting relevant research on AcWRKY28, your suggestion makes us aware that there may be something wrong in our research. In order to ensure the accuracy and scientific of the research results, we checked all experimental materials and results. We found that the study gene was Aco000358.1, and this gene also named AcWRKY31 based on Xie et al. paper. Due to the wrong labeling of plant material names in our early sequencing process, we provided the wrong protein sequence diagram. After a series of checks, we found that there was no problem with other experimental results except that the protein sequence diagram was wrong. Thank you very much for your contribution to this study. You put forward a very important suggestion for us, so that we found this error in time and made corrections. At the same time, we are also very sorry for this mistake.

Point 2: The pineapple WRKY TF family has 54 members. The authors do not justify the choice of the gene they have decided to study. If one looks at the Xie et al. paper again (fig. 7c), the AcWRKY28 and AcWRKY50 genes are reported to be the most induced by cold. Whatever the motivation behind the choice of the gene, it has to be stated.

Response 2: Thank you for your professional comments. It has been reported that the expression of AcWRKY31 was induced under cold and drought stress condition, but the function of AcWRKY31 in still limited. Due to the immature genetic transformation system of pineapple, it is still difficult to study the function of AcWRKY31 in pineapple. Since both rice and Arabidopsis are important model plants, we constructed AcWRKY31 overexpression transgenic plants to further investigate the function of AcWRKY31. We have explained our motivation behind the choice of AcWRKY31 in the revised version (Line 111-122).

Point 3: The nuclear localization of AcWRKY28 in pineapple cells has already been demonstrated in the following paper by some of the co-authors: Priyadarshani et al. (2019) Biomolecules 9, 617. Therefore, this paper should be cited and the nuclear localization experiment in tobacco is no longer relevant and should be deleted.

Response 3: Thank you for your suggestion. In the current study, in order to avoid this problem, we searched the relevant reports of AcWRKY31.We could not find any information about the subcellular localization of AcWRKY31, so we retained this part of the results.

Point 4:  To demonstrate the presence of a transcription activation domain in the WRKY28 protein, the authors fused its coding sequence to the yeast Gal DNA-binding domain coding sequence of pGBKT7. It then makes no sense to transform yeast with both this plasmid and pGADT7-T which contains the Gal activation domain. Only pGBKT7-AcWRKY28 and pGBKT7 as a control should have been used in this experiment.

Response 4: Thank you for your professional suggestion. We have revised this figure and the related sentences in the revised version based on you suggestion (Figure 2B, Line 148-305 and Line 658-664).

Point 5: The expression profiles of the pineapple AcWRKY28 gene after stresses show some kind of "up and down" patterns that are very reminiscent of the diurnal variations described in the following paper (that should be cited): Sherma et al. (2017) Diurnal cycling transcription factors of pineapple revealed by genome-wide annotation and global transcriptomic analysis. Genome Biol. Evol. 9:2170–2190. According to this work, AcWRKY28 shows a diurnal cycling of expression. The lack of induction of expression of AcWRKY28 by cold should be discussed in relation with the data published by Xie et al. (fig 7c).

Response 5: Thank you for your valuable suggestion. In the revised version, we described the expression trend of AcWRKY31 based on the paper of Sherma et al. (2017 ), and we also added the introduction of AcWRKY31 expression published by Xie et al. (2018) (Line 111-113 and Line 539-541 )

Point 6:  There is little information about the rice and Arabidopsis AcWRKY28-overexpressing lines used here: which promoters were used to drive the expression? Were the levels of expression different between the two lines used per species? Were the AcWRKY28 level of expression correlated with the level of expression of some of the effector genes? The details of the constructs and the transgene expression data should be added.

Response 6: Thank you for your suggestion. In our study, the expression of AcWRKY31 was driven by CaMV 35S promoter, and we have added this information in the revised version (Line 140-142 and Line 650-651). We performed qRT-PCR to investigate the expression of AcWRKY31 in wild-type and transgenic plants. The results showed that the expression levels of the two lines per species are different (Figure S1). We also found that the transgenic plants with higher AcWRKY31 expression level were more sensitive to drought and salt stress (Figure 5-8).

Point 7:  From this work, it is difficult to draw any conclusion about the function of the AcWRKY28 gene in pineapple. Yes, its overexpression reduced the tolerance of heterologous transgenic plants to various stresses but there could be several possible explanations for this. Some WRKY TF are known to act as negative feedback regulators of MPK (Chen et al. (2019) WRKY transcription factors: evolution, binding, and action. Phytopathology Research 1:13). The TF genes are generally expressed at low level. The abnormal abundance of a TF in overexpressing plants could result in the disruption of pathways involving other endogenous TF. These disruptions could explain the decrease of stress resistance in overexpressing plants although this mode of action might not be relevant in plants producing the TF at a normal level. Similarly, some WRKY TF are known to interact with kinases. One can easily imagine how an abnormal abundance of a TF in overexpressing plants could disrupt endogenous pathways by acting as a "trap" for TF-interacting kinases. Therefore, the conclusion (line 183-184) "These results suggest that AcWRKY50 serves as a negative regulator of plant growth and development under drought and salt stresses" should be mitigated because there is no direct evidence for this from the data. The evidence could only be brought up by the study of an AcWRKY28 KO pineapple plant.

Response 7: We sincerely appreciate for your professional suggestion. WRKY TFs participates in a regulatory network that integrates internal and environmental factors to regulate various stress responses. Since the function of WRKY TFs is complex, we mitigate this conclusion and revised the related sentences in the revised version.

Point 8: The supplementary table 2 is of little use without any gene function mentioned. There should be an extra column indicating the putative function of each gene.

Response 8: Thank you for your professional suggestion. We have added the putative function of each gene in the Supplementary Table 3.

Point 9:  The quality of the writing is generally good but a few sentences need editing. Here are two examples from the abstract:

- line 19: ...overexpression transgenic rice plants WERE significantly reduced

- line 21-22: ectopic overexpression of AcWRKY50 in rice and Arabidopsis exhibited RESULTED IN PLANT OVER sensitivity to drought and salt stress ES

Response 9: We are very sorry for this mistake in the manuscript, we have revised these sentences in the revised version.

Round 2

Reviewer 3 Report

The authors have addressed the issues of their version 1, except for the lack of functions of the up and down regulated genes of table S2. Instead of adding the gene functions in an extra column in table S2 as suggested, they have created a new table (S3) containing only the gene functions. This does not make table S2 more useful. Besides, tables S2 and S3 do not contain the same number of entries. Table S3 is of no use and the gene functions still have to be added to table S2. This would allow the user to directly correlate gene regulation and gene function.